# RTTOV-gb v1.0 - Updates on sensors, absorption models, uncertainty, and availability

Domenico Cimini[1,2], James Hocking[3], Francesco De Angelis[2], Angela Cersosimo[1], Francesco Di Paola[1], Donatello Gallucci[1], Sabrina Gentile[1], Edoardo Geraldi[1], Salvatore Larosa[1], Saverio Nilo[1], Filomena Romano[1], Elisabetta Ricciardelli[1], Ermann Ripepi[1], Mariassunta Viggiano[1], Lorenzo Luini[4], Carlo Riva[4], Frank S. Marzano[5,2], Pauline Martinet[6], YunYoung Song[7], Myoung Hwan Ahn[7], and Philip W. Rosenkranz[8].

[1]National Research Council of Italy, Institute of Methodologies for Environmental Analysis, Potenza, 85050, Italy

[2]Center of Excellence CETEMPS, University of L'Aquila, L'Aquila, 67100, Italy

[3]MET OFFICE, Exeter, United Kingdom

[4]DEIB - Politecnico di Milano, IEIIT – CNR, Milano, Italy

[5]University of Rome La Sapienza, Rome, Italy

[6]CNRM, Université de Toulouse, Météo-France, CNRS, Toulouse, France

[7]School of Engineering, Ewha Womans University, Seoul, South Korea

[8]Massachusetts Institute of Technology, Cambridge, MA, 02139, USA

*Correspondence to:* D. Cimini (domenico.cimini@imaa.cnr.it)

**Abstract.** This paper describes the first official release (v1.0) of RTTOV-gb. RTTOV-gb is a FORTRAN 90 code developed by adapting the atmospheric radiative transfer code RTTOV, focused on satellite observing geometry, to the ground-based observing geometry. RTTOV-gb is designed to simulate ground-based upward-looking microwave radiometer (MWR) observations of atmospheric downwelling natural radiation in the frequency range from 22 to 150 GHz. Given an atmospheric profile of temperature, water vapour and, optionally, cloud liquid water content, and together with a viewing geometry, RTTOV-gb computes the downwelling radiances and brightness temperatures leaving the bottom of atmosphere in each of the channels of the sensor being simulated. In addition, it provides the sensitivity of observations to the atmospheric thermodynamical state, i.e. the Jacobians. Therefore, RTTOV-gb represents the forward model needed to assimilate ground-based MWR data into numerical weather prediction models, which is currently pursued internationally by several weather services. RTTOV-gb is fully described in a previous paper (De Angelis et al., 2016), while several updates are described here. In particular, two new MWR types and a new parameterization for atmospheric absorption model have been introduced since the first paper. In addition, estimates of the uncertainty associated with the absorption model and with the fast parameterization are given here. Brightness temperatures ($T_B$) computed with RTTOV-gb v1.0 from radiosonde profiles have been compared with ground-based MWR observations at six channels (23.8, 31.4, 72.5, 82.5, 90.0, and 150.0 GHz). The comparison shows statistics within the expected accuracy. RTTOV-gb is now available to licensed users free of charge from the Numerical Weather Prediction Satellite Application Facility (NWP SAF) website, after registration. Coefficients for four MWR instrument types and two absorption model parameterizations are also freely available from the RTTOV-gb support website.

## 1   Introduction

RTTOV-gb is a fast radiative transfer code, designed to simulate ground-based upward-looking microwave radiometer (MWR) observations of atmospheric downwelling natural radiation (i.e. radiances). RTTOV-gb consists in the FORTRAN-90 code described by De Angelis et al. (2016), developed by adapting version 11.2 of RTTOV, the Radiative Transfer for the TIROS Operational Vertical Sounder (TOVS), which is designed to simulate the satellite observation perspective only. From its first implementation (Eyre, 1991) through to its current version (Saunders et al., 2018), RTTOV simulates radiances from space-borne passive sensors, and also computes the Jacobians, i.e. the gradient of the radiances with respect to the atmospheric state vector. RTTOV is widely used by many national and international meteorological services for assimilating down-looking observations from visible, infrared, and microwave radiometers, spectrometers and interferometers aboard satellite platforms. For this reason, RTTOV is maintained and continuously developed by the Numerical Weather Prediction (NWP) Satellite Application Facility (SAF) of the European Organization for the Exploitation of Meteorological Satellites (EUMETSAT). However, satellite passive observations are known to lack accuracy and resolution in the planetary boundary layer (PBL), leaving a so-called observational gap between surface and upper troposphere (National Research Council, 2008). Therefore, in the last decade there has been increasing interest for ground-based sensors that could help bridging the PBL observational gap (Illingworth et al., 2015; Illingworth et al., 2019), including ground-based microwave radiometers (MWR). Ground-based MWR observations are also widely used for radiopropagation studies and the characterization of atmospheric attenuation for telecommunication channels (Riva et al., 2014).

Data assimilation of MWR observations into NWP models may be particularly important in forecasting weather and atmospheric attenuation. In order to assimilate ground-based radiometric observations, namely brightness temperatures ($T_B$), a fast radiative transfer forward model is needed. This model allows rapid simulations of $T_B$ at selected radiometer channels based on the NWP model state vector, i.e. atmospheric temperature and humidity profiles, similar to what RTTOV does for satellite sensors. Therefore, in the framework of the COST Actions EG-CLIMET and TOPROF, there have been continuous activities to develop a ground-based version of RTTOV: RTTOV-gb (De Angelis et al., 2016). RTTOV-gb is a one-dimensional radiative transfer model: for any given location, it takes vertical profiles of atmospheric temperature, water vapour, and cloud liquid water specified on an arbitrary set of pressure levels, and from them it simulates $T_B$ as well as the Jacobians (i.e. sensitivity of $T_B$ to atmospheric thermodynamical profiles) corresponding to ground-based upward-looking microwave radiometers. The radiative transfer extends to the top of the atmosphere and it does include extra-terrestrial contribution, i.e. the cosmic background radiation. The availability of RTTOV-gb is fostering wider use of MWR observations in NWP models, as demonstrated by the current use at some of the most relevant meteorological services in Europe as well as outside, such as Météo-France, the German Meteorological Service (Deutscher Wetterdienst, DWD), Korean Meteorological Administration (KMA).

This paper introduces several updates of RTTOV-gb since its first development (De Angelis et al., 2016). In section 2 we introduce a new absorption model parameterization and two new sensors that have been added among the setting options. Section 3 presents the evaluation of RTTOV-gb against the reference line-by-line

1 radiative transfer model and real radiometric ground-based observations. Section 4 summarizes the findings
2 while Section 5 provides instructions for code and data access and use.

## 4   2   RTTOV-gb updates

### 5   2.1   New sensors

Similar to RTTOV, RTTOV-gb was designed to simulate observations and Jacobians for a suite of instruments,
in this case ground-based instead of satellite-borne sensors. While introducing RTTOV-gb, De Angelis et al.
(2016) presented results for two sensors, among the most common ground-based MWR worldwide: the
Humidity And Temperature PROfiler (HATPRO) manufactured by RPG and the MP3000A manufactured by
Radiometrics. In the current version (v1.0), two more sensors have been added to the suite: the microwave
temperature radiometer TEMPERA (Stähli et al., 2013; Navas-Guzmán et al., 2017) and the Liquid Water Path
(LWP) K-to-W-band radiometer (LWP_K2W). Note that LWP_K2W is a virtual instrument which includes all
the channels offered by the LWP family of ground-based radiometers (LWP, LWP-U90, LWP-U72-82, LWP-
U150, LWP-90-150) manufactured by RPG (https://www.radiometer-physics.de/products/microwave-remote-
sensing-instruments/radiometers/lwp-radiometers/ ).

The RTTOV-gb optical depth calculation is a parameterisation which requires pre-computed coefficients. These
coefficients are specific to each instrument and are stored in coefficient files (see also Section 2.2). Every time a
new sensor is added to the sensor suite, a dedicated coefficient file must be generated. The coefficient file
contains the regression coefficients to estimate the optical depth for each atmospheric layer and each sensor
channel from the thermodynamical properties of the layer through a set of predictors. The predictors are derived
from the input state vector profile and depend on the elevation angle $\theta$ and pressure P, temperature T, and
specific humidity Q at the considered and surrounding levels. The regression coefficients are trained on a set of
diverse profiles which covers atmospheric conditions of different climate zones. Pressure levels and regression
limits for T and Q are reported in Table 1. The coefficients are based on a set of 101 pressure levels specifically
created for RTTOV-gb which are more dense in the lower atmosphere than the RTTOV coefficient levels
usually used for space-borne sensors.

The list of currently supported sensors with their channel frequencies is given in Table 2. Other sensors are
planned for future updates, e.g., those operating at 183 GHz for low water vapor and cloud liquid water retrievals
(Cimini et al., 2007).

### 31   2.2   Absorption model

Similar to RTTOV, RTTOV-gb is a parametrized atmospheric radiative transfer code. In the microwave region
and for clear sky conditions, the parameterization only affects the atmospheric gas absorption. This means that
the optical depth of each layer is only due to absorption by atmospheric gases (mainly oxygen, water vapor, and
nitrogen). The parameterization consists in the fact that the layer optical depth is not computed from a complex

line-by-line (LBL) absorption model (Clough et al. 2005), but rather from a simplified parametrized model. The simplified model consists in a linear regression, which relates the layer optical depth to predictors derived from the layer atmospheric thermodynamical properties (i.e. pressure, temperature, and humidity). The regression coefficients are computed off-line from a diverse training dataset of atmospheric thermodynamical profiles and corresponding optical depths computed with a LBL model. Thus, RTTOV-gb provides a fast parameterization of the LBL model adopted for the training of the regression coefficients. For the microwave frequency range (10–200 GHz), the regression coefficients of RTTOV are trained using the AMSUTRAN LBL model developed at the Met Office (Turner et al., 2018) which is based on the millimeter-wave propagation model (MPM) introduced by Liebe (1989), with some modifications following Tretyakov et al. (2005), Liljegren et al. (2005), and Payne et al. (2008) (Saunders et al. 2017). Conversely, RTTOV-gb was trained using a later version of MPM, described by Rosenkranz (1998, hereafter R98), which is probably the most used among the ground-based microwave radiometry community. This model is continuously revised and freely available (Rosenkranz, 2017 hereafter R17), and its uncertainty has been carefully investigated (Cimini et al., 2018). Therefore, RTTOV-gb has been trained using the R17 model also (version of 17/05/2017 available at http://cetemps.aquila.infn.it/mwrnet/lblmrt_ns.html). Coefficients for both R98 and R17 models are now available within RTTOV-gb v1.0. Extending the results in Cimini et al. (2018) from 60 to 150 GHz, Figure 1 shows clear-sky zenith downwelling $T_B$ computed with R17 model and the difference between $T_B$ computed with the two model versions, for six reference atmosphere climatology conditions. The difference spans from -2 to +3 K in the considered frequency range and thus it is not negligible for the sensors currently available for RTTOV-gb v1.0.

As mentioned, Cimini et al. (2018) investigated the uncertainty of $T_B$ computed with R17 model due to the laboratory uncertainty of the adopted spectroscopic parameters. Through a sensitivity test, they identified 111 parameters (6 for water vapor and 105 for oxygen), whose contribution to the total uncertainty was dominant with respect to others. For these 111 parameters, Cimini et al. (2018) estimated the full uncertainty covariance matrix ($\mathbf{Cov}(\boldsymbol{p})$), from which the $T_B$ uncertainty covariance matrix ($\mathbf{Cov}(\boldsymbol{T_B})$) and the square root of its diagonal terms ($\boldsymbol{\sigma}(\boldsymbol{T_B})$) were computed. $\boldsymbol{\sigma}(\boldsymbol{T_B})$ represents the standard deviation of typical spectroscopic uncertainties to be expected from $T_B$ computed with R17 model. Figure 2 shows $\boldsymbol{\sigma}(\boldsymbol{T_B})$ for zenith observations in six climatological atmospheric conditions. Note that uncertainties used here are at 1-sigma level, i.e. applying an unitary coverage factor (k=1, as defined by JCGM, 2008).

Note that the analysis of Cimini et al. (2018) was limited to the 20-60 GHz range. Here, a new sensitivity analysis has been performed to cover the frequency range of sensors available for RTTOV-gb v1.0 (20 to 150 GHz). One additional parameter was found to contribute dominantly, namely the temperature-dependence exponent $n_{cs}$ of the water vapor self-broadened continuum, contributing with its uncertainty by 0.2-0.6 K to the total uncertainty of downwelling $T_B$ between 70-150 GHz. By applying the same approach as described in Cimini et al. (2018) for other water vapor continuum parameters, the covariance and correlation between $n_{cs}$ and the self-broadened continuum parameter $C_s$ were estimated to be $\mathbf{Cov}(C_s,n_{cs})$=-3.6208e$^{-10}$ (km$^{-1}$ hPa$^{-2}$ GHz$^{-2}$) and $\mathbf{Cor}(C_s,n_{cs})$=-0.183, respectively. The covariance of $n_{cs}$ with respect to the other 111 parameters is estimated to be negligible.

For more details on RTTOV and the differences between RTTOV-gb and RTTOV, see Hocking et al. (2015), Saunders et al. (2018), and De Angelis et al. (2016).

## 3 Validation with reference model and real observations

The accuracy of RTTOV-gb v1.0 $T_B$ simulations has been tested against both the reference LBL model and real ground-based observations.

### 3.1 Validation against reference model

As described by De Angelis et al. (2016), the approach for testing RTTOV-gb against the reference LBL model used for training (i.e. R98 or R17) consists in computing $T_B$ simulations with both models from a set of independent profiles (i.e., not used for training) and to evaluate the statistics of their difference, namely the mean (bias) and root-mean-square (rms) difference. For the original two sensors (HATPRO and MP3000A), De Angelis et al. (2016) reported in their Tables 2 and 3 the statistics (bias and rms) for the comparison between RTTOV-gb and the LBL model used for training (R98 in their case) against an independent profile set at four elevation angles (90, 30, 19, and 10°). Similarly, here we report the statistics for the two new sensors (i.e. TEMPERA and LWP_K2W) and the same R98 LBL model, respectively in Table 3 and 4. These two tables show that the discrepancies between RTTOV-gb v1.0 and LBL optical depths lead to negligible $T_B$ differences. The rms differences at zenith are lower than 0.18 K for all channels. When decreasing the elevation angle, the rms differences generally decrease for 50-57 GHz channels (Table 3), while they increase for 23/31 and 70-150 GHz channels (Table 4), in accordance with the different atmospheric opacity. The highest rms differences (0.3 K) are found for window channels 31 and 150 GHz at 10° elevation. Similarly to De Angelis et al. (2016), the main conclusion is that the uncertainty introduced by the fast model approximation (RTTOV-gb) is within the typical instrument uncertainty and thus does not dominate the uncertainty budget of observations vs. simulations. Let us underline that Tables 2 and 3 of De Angelis et al. (2016) and Tables 3 and 4 of this paper report statistics when using R98 LBL model for training. The analogous rms obtained using the LBL model R17 at zenith are reported in Table 5 as "fast parameterization uncertainty". As expected, rms values do not differ significantly from those obtained against R98. In fact, this test only tells about the accuracy of the parametrized regression in reproducing the LBL model radiances, which is largely independent of the choice of the LBL model. Table 5 also reports the $T_B$ uncertainty contribution due to the uncertainty of spectroscopic parameters (from Figure 2). The estimated total uncertainty is computed as the sum in quadrature (i.e. the square root of the sum of squares) of two contributions: the uncertainty due to fast parameterization and absorption model spectroscopic parameters. The latter dominates the uncertainty budget. The total uncertainty so estimated is reported in Table 5 for each sensor and channel available in RTTOV-gb.

### 3.2 Validation against real observations

RTTOV-gb $T_B$ simulations have been previously compared with real ground-based observations from six HATPRO and one MP3000-A (De Angelis et al., 2016; 2017). The frequency range covered by HATPRO and MP3000-A channels overlaps the frequency range of TEMPERA, so we assume RTTOV-gb has been tested for this sensor as well. Conversely, the frequency range of LWP_K2W extends to higher frequencies (up to 150 GHz) to include all the channels offered by the RPG LWP ground-based radiometer family (LWP, LWP-U90,

LWP-U72-82, LWP-U150, LWP-90-150). Thus, in the following we present a comparison with observations from a LWP-U72-82 radiometer located at the Polytechnic University campus in Milan (Italy, 45.450 N, 9.183 E), and from a LWP-90-150 radiometer located at the Atmospheric Radiation Measurement (ARM) program Southern Great Plains (SGP) central facility in Lamont (OK, USA, 36.605 N, 97.485 W). The two datasets represent respectively midlatitude summer and midlatitude winter conditions. Operational radiosondes are launched from these two sites, measuring profiles of pressure, temperature, and water vapor, which are then processed by RTTOV-gb to compute downwelling $T_B$ simulating ground-based observations. Since these radiosondes do not measure liquid water content profiles, we assume no clouds are present and thus meaningful comparison with real observations can be performed in clear sky only. Radiosondes usually reach up to 10 hPa (~30 km altitude), leaving the five uppermost RTTOV-gb levels to be covered with climatological profiles. This has negligible impact on ground-based radiance calculations.

The LWP-U72-82 instrument has four channels (23.84, 31.4, 72.5, and 82.5 GHz) and it is mainly used for radiopropagation studies. The available dataset extends for one month (from 16 June to 15 July 2018), corresponding to relatively moist midlatitude summer conditions. The dataset includes radiometric observations and pressure, temperature, and humidity profiles measured by radiosonde ascents launched twice-daily from the Milan Linate airport (~20 km from the Politechnic University campus). Radiometric observations are collected at a fixed elevation angle (35.3°), matching the direction of the Alphasat telecommunication link. Absolute MWR calibrations were performed nine months earlier and four months later than the period under study, showing no substantial change in the calibration coefficients. Thus, we assume the calibration was stable during the period under study. An example of data is shown in Figure 3 for three consecutive days. Here, $T_B$ observed at the four channels are plotted together with RTTOV-gb simulations and their estimated uncertainty. It appears that simulations usually fit the observations within uncertainty, except for periods with clouds (at ~167.0, i.e. 00:00 of June 16) and rain (~169.0, i.e. 00:00 of June 18). This is expected as RTTOV-gb simulations are computed from radiosonde measurements, which do not provide hydrometeor content, and thus do not take into account the radiative contribution of clouds and rain. Thus, for a fair clear-sky comparison, data affected by either rain or clouds must be screened out. As illustrated in Figure 3, the LWP-U72-82 is equipped with a rain sensor, indicating either rain or no-rain on the antenna radome. Observations during rain, as flagged by the rain sensor, have been discarded. In addition, cloudy conditions have been identified by setting a threshold on the standard deviation of $T_B$(31.4GHz) over a time period (Turner et al. 2007). This approach has been used previously with a 0.5 K threshold over 1-hour period (De Angelis et al. 2017). Here, we choose a shorter period (10 minutes); assuming lower clear-sky atmospheric variability with decreasing time interval, and looking at the distribution of $T_B$(31.4GHz) standard deviation, we set the threshold to 0.2 K. Thus, data identified as cloudy, by the standard deviation of $T_B$(31.4GHz) over a 10-minute period being larger than 0.2 K, have been discarded. The cloud and rain screening reduced the dataset by ~33%, leaving 40 match-ups between clear sky radiosonde and radiometric observations (averaged within ±5 minutes from the radiosonde launch). Scatter plots of simulated vs. observed $T_B$ at 35.3° elevation for the four channels of LWP-U72-82 are shown in Figure 4. Note that the correlation coefficient is 0.98 for all four channels. The slope is within 5% for all channels but 72.5 GHz (~8%), for which the difference between observations and simulations tend to increase as $T_B$ decrease. This may be due to an issue with the instrument gain calibration, as well as to increasing uncertainty for this channel at lower temperature

and moisture conditions (see Figure 2). Statistics at 23.84 and 31.4 GHz are of the same magnitude of those reported by De Angelis et al. (2017) at 30° elevation (their Figure 5, panel C).

The LWP-90-150 instrument has two channels (90.0 and 150.0 GHz) and it is mainly used for the retrieval of total column cloud liquid water content. The instrument considered here has been running at the ARM SGP central facility between November 2006 and November 2013, performing calibrations regularly using the tip curve method (Cadeddu et al., 2013). Here we exploit a 2-month dataset of radiometric and radiosonde observations (ARM, 2018a & 2018b) collected in January-February 2012. This dataset corresponds to relatively dry midlatitude winter conditions. An example of data is shown in Figure 5 for three consecutive days, corresponding to a dry clear-sky period with intermittent thick clouds and rain. Observations flagged by the rain sensor have been discarded. In addition, cloudy conditions have been identified with the same approach as described above, i.e. setting a threshold on the 10-min standard deviation of $T_B$ at a window channel, here replacing the 31.4 GHz with the 90.0 GHz channel. However, since $T_B$(90GHz) has ~6 times larger sensitivity to water vapor (Cimini et al., 2007), the clear-sky threshold is increased by the same factor, i.e. 1.2 K. Thus, data with 10-minute standard deviation of $T_B$(90GHz) larger than 1.2 K have been discarded. The cloud and rain screening reduced the dataset by ~26%, leaving 173 match-ups between clear sky radiosonde and radiometric observations (averaged within ±5 minutes from the radiosonde launch). Scatter plots of simulated vs. observed $T_B$ at 90° elevation for the two channels of LWP-90-150 are shown in Figure 6. The correlation coefficient is 0.95 and 0.99 for 90 and 150 GHz, respectively, while the slope is within 4% for both channels.

Overall, the average differences at all the six LWP_K2W channels are close to the accuracy estimated in Table 5D. A direct comparison is given in Figure 7. Here, the estimated uncertainty for the six LWP_K2W channels is compared with the experimental mean difference between simulations and observations. Note that radiometric observations at the four lower frequency channels were collected in June-July in Milan (45°N), while in January-February in Lamont (36°N) at the two higher frequency channels. Thus, the simulation uncertainty is estimated using midlatitude summer conditions for the four lower frequency channels, while midlatitude winter conditions for the two higher frequency channels. The experimental bias is generally larger than the simulation estimated uncertainty, as one would expect since the observations are also affected by uncertainty. Except for the 72.5 GHz channel, the estimated uncertainty and experimental bias are within 0.5 K, which corresponds to the absolute $T_B$ accuracy claimed by the manufacturer for the LWP radiometer series. At 72.5 GHz, as anticipated, observations-simulations differences tend to increase as $T_B$ decrease, possibly due to either conditions-dependent uncertainty or an issue with the instrument gain calibration. This will be subject of future investigation.

## 4    Summary and future developments

RTTOV-gb v1.0 is now freely available, after website registration (see Section 5). The updates with respect to the original development (described in De Angelis et al., 2016) are presented here, including two additional sensors, an additional parameterization for the training atmospheric absorption model, and an estimate of the $T_B$ uncertainty.

RTTOV-gb v1.0 has been trained and validated against two versions of a reference line-by-line absorption model, i.e. R98 (Rosenkranz, 1998) and R17 (Rosenkranz, 2017). In the frequency range commonly covered by RTTOV-gb v1.0 sensors, $T_B$ rms differences are smaller than typical sensor uncertainties at all considered

channels and for both the reference absorption models. $T_B$ computed with RTTOV-gb v1.0 from radiosonde profiles have been compared with simultaneous ground-based radiometric observations at six channels (23.84, 31.4, 72.5, 82.5, 90.0, and 150.0 GHz) and two observing elevation angle (35.3° and 90°). Differences between simulated and measured $T_B$ are within uncertainty as expected from instrumental and simulation contributions. Future developments include additional sensors (e.g., at 183 GHz) and characterization of liquid water absorption uncertainties.

We expect this paper will provide a reference for the exploitation of RTTOV-gb for MWR data assimilation into NWP models, as already started at some meteorological services in Europe as well as in other continents.

## 5 Code and data availability

RTTOV-gb v1.0 is available to licensed users free of charge. RTTOV-gb may be obtained by registering (https://www.nwpsaf.eu/site/register/) with the NWP SAF website (https://www.nwpsaf.eu/) and then selecting RTTOV-gb in your software preferences. Instructions for compiling and running RTTOV-gb are provided in the RTTOV-gb User Guide within the software package. The software package also includes scripts to verify the installation and FORTRAN code examples for running the RTTOV-gb forward and K (Jacobian) modules. RTTOV-gb is designed for UNIX/Linux systems. The software is now successfully tested on the following architectures and Fortran 90 compilers: Intel systems with gfortran, ifort, NAG, and pgf90, and Apple Mac systems with gfortran.

The RTTOV-gb v1.0 code is based on RTTOV v11.2 and the programming interface is identical to that version of RTTOV, though some inputs and outputs are not used by RTTOV-gb. The original RTTOV v11.2 can be obtained from the NWP SAF web site (http://nwpsaf.eu/site/software/rttov/rttov-v11/). Thus, the computational performances of RTTOV-gb is similar to that of RTTOV v11.2, which have been reported (https://www.nwpsaf.eu/site/download/documentation/rtm/docs_rttov11/Performance_Tests_RTTOV_v11.2.pdf ). For clear-sky microwave simulations, the main factor in simulation speed is the number of coefficient levels, which is 101 for RTTOV-gb. Typical clear-sky run-times for RTTOV-gb are ~0.25 ms per profile for the direct model and ~1.0 ms per profile for the Jacobian model, though timings are dependent on the hardware, compiler, and compiler flags being used, as well as, for example, the number of levels in the input profile, the number of channels simulated per profile, and the inclusion or not of cloud liquid water.

Note that RTTOV-gb is not supported by NWP SAF. All questions, bug reports or requests for new coefficients should be sent to rttovgb@aquila.infn.it. Always refer to the RTTOV-gb web page for bug fixes, new coefficients, and code updates: http://cetemps.aquila.infn.it/rttovgb/rttovgb.html.

The RTTOV-gb package contains optical depth coefficient files for sensors supported by RTTOV-gb at the time of release. Coefficients for sensors not currently considered can be requested to rttovgb@aquila.infn.it. Note that RTTOV-gb only supports microwave sensors currently. Other resources include:

- Default pressure levels: http://cetemps.aquila.infn.it/mwrnet/main_files/DAT/RTTOVgb_101_levels_p.dat
- Regression coefficients: http://cetemps.aquila.infn.it/mwrnet/rttovgb_coefficients.html

- Regression limits: http://cetemps.aquila.infn.it/mwrnet/main_files/DAT/RTTOVgb_101_pressure_levels_and_regression_limits.xlsx

- NWP SAF profile sets used for the RTTOV-gb training and independent test: https://nwpsaf.eu/deliverables/rtm/profile_datasets.html.

For more information on reference profiles and regression limits see the related link on the official RTTOV website (https://www.nwpsaf.eu/site/software/rttov/download/coefficients/coefficient-download/#Reference_profiles_and_regression_limits ).

Finally, the absorption model by Rosenkranz (2017) is available as a FORTRAN 77 code at http://doi.org/10.21982/M81013. Older versions, including the one used here (2017/05/15), are available at http://cetemps.aquila.infn.it/mwrnet/lblmrt_ns.html.

## 6    Acknowledgements

This work has been stimulated through COST Actions supported by COST (European Cooperation in Science and Technology). Support from the European Space Agency through the WRad project (ESA Contract No. 4000125141/18/NL/AF) is acknowledged. Part of this research was funded by the Korea Meteorological Administration Research and Development Program under Grant KMI2018-07410. Microwave radiometer and radiosonde data in Lamont (OK, USA) were obtained from the Atmospheric Radiation Measurement (ARM) User Facility, a U.S. Department of Energy (DOE) Office of Science user facility managed by the Office of Biological and Environmental Research.

*Author contributions*. DC, JH, and FDA designed the research, contributed to data processing and analysis, and wrote the original manuscript. PWR, PM, YS, and MHA contributed to the investigation in Section 2. LL, CR, FSM contributed with curation of observed data. FDP, DG, SG, FR, ER, and ER contributed with software development. AC, EG, SL, SN, MV contributed to validation data analysis. All the co-authors helped to revise the manuscript.

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

| Level (#) | Pressure (1e3 hPa) | Minimum T (K) | Maximum T (K) | Minimum Q (ppmv) | Maximum Q (ppmv) |
|---|---|---|---|---|---|
| 1 | 0.0000 | 143,65 | 245,95 | 9,1330E-01 | 5,2410E+00 |
| 11 | 0.0379 | 162,77 | 279,05 | 1,3280E+00 | 6,0170E+00 |
| 21 | 0.1349 | 169,71 | 259,26 | 1,2860E-02 | 1,0250E+02 |
| 31 | 0.2700 | 182,27 | 278,60 | 1,2860E-02 | 4,5660E+03 |
| 41 | 0.4251 | 195,91 | 303,26 | 2,3870E+00 | 1,6690E+04 |
| 51 | 0.5841 | 196,73 | 315,57 | 4,8630E+00 | 2,8090E+04 |
| 61 | 0.7336 | 189,96 | 332,20 | 8,8570E+00 | 3,7010E+04 |
| 71 | 0.8624 | 189,96 | 342,43 | 7,5350E+00 | 4,4160E+04 |
| 81 | 0.9618 | 189,96 | 349,92 | 6,7550E+00 | 5,1280E+04 |
| 91 | 1.0256 | 189,96 | 350,08 | 6,3350E+00 | 4,7540E+04 |
| 101 | 1.0500 | 189,96 | 350,08 | 6,1880E+00 | 4,7640E+04 |

---

[1]http://cetemps.aquila.infn.it/mwrnet/main_files/DAT/RTTOVgb_101_pressure_levels_and_
regression_limits.xlsx

1     **Table 2**: Sensors supported by RTTOV-gb as for October 2018 (v1.0) with the corresponding number

2     of channels and their central frequency.

| Sensor | RTTOV-gb ID | Number of channels (#) | Channel frequencies (GHz) |
|--------|-------------|------------------------|---------------------------|
| HATPRO | 1 | 14 | 22.24; 23.04; 23.84; 25.44; 26.24; 27.84; 31.40; 51.26; 52.28; 53.86; 54.94; 56.66; 57.30; 58.00 |
| MP3000A | 2 | 22 | 22.234; 22.500; 23.034; 23.834; 25.000; 26.234; 28.000; 30.000; 51.248; 51.760; 52.280; 52.804; 53.336; 53.848; 54.400; 54.940; 55.500; 56.020; 56.660; 57.288; 57.964; 58.800; |
| TEMPERA | 3 | 12 | 51.25; 51.75; 52.25; 52.85; 53.35; 53.85; 54.40; 54.90; 55.40; 56.00; 56.50; 57.00 |
| LWP_K2W | 4 | 6 | 23.84; 31.40; 72.50; 82.50; 90.0; 150.0 |

**Table 3**: Statistics for the comparison between RTTOV-gb and the line-by-line model R98 (Rosenkranz, 1998) used for training against an independent profile set. The TEMPERA instrument channel number (Chan no.), the channel central frequency, bias, and rms at four elevation angles are reported.

| Chan no. (#) | Central frequency (GHz) | Bias (K) | | | | rms (K) | | | |
|---|---|---|---|---|---|---|---|---|---|
| | | 90° | 30° | 19° | 10° | 90° | 30° | 19° | 10° |
| 1 | 51.250 | -0.003 | -0.019 | -0.018 | -0.043 | 0.153 | 0.158 | 0.125 | 0.077 |
| 2 | 51.750 | -0.003 | -0.016 | -0.012 | -0.031 | 0.160 | 0.148 | 0.104 | 0.049 |
| 3 | 52.250 | -0.004 | -0.010 | -0.006 | -0.020 | 0.167 | 0.131 | 0.077 | 0.029 |
| 4 | 52.850 | -0.003 | 0.001 | -0.002 | -0.010 | 0.165 | 0.093 | 0.041 | 0.019 |
| 5 | 53.350 | -0.001 | 0.006 | -0.003 | -0.004 | 0.141 | 0.054 | 0.021 | 0.015 |
| 6 | 53.850 | -0.001 | 0.002 | -0.001 | -0.002 | 0.095 | 0.026 | 0.015 | 0.012 |
| 7 | 54.400 | 0.001 | -0.002 | -0.001 | -0.001 | 0.047 | 0.015 | 0.011 | 0.007 |
| 8 | 54.900 | 0.002 | 0.000 | -0.000 | -0.000 | 0.024 | 0.011 | 0.008 | 0.004 |
| 9 | 55.400 | 0.002 | 0.001 | 0.000 | -0.000 | 0.017 | 0.008 | 0.005 | 0.002 |
| 10 | 56.000 | 0.003 | 0.000 | 0.000 | 0.000 | 0.013 | 0.005 | 0.003 | 0.001 |
| 11 | 56.500 | 0.002 | 0.001 | 0.000 | 0.000 | 0.011 | 0.004 | 0.002 | 0.001 |
| 12 | 57.000 | 0.002 | 0.000 | 0.000 | 0.000 | 0.009 | 0.003 | 0.001 | 0.000 |

1    **Table 4**: Same as Table 3 but for the LWP_K2W instrument.

| Chan no. (#) | Central frequency (GHz) | Bias (K) | | | | rms (K) | | | |
|---|---|---|---|---|---|---|---|---|---|
| | | 90° | 30° | 19° | 10° | 90° | 30° | 19° | 10° |
| 1 | 23.840 | 0.008 | 0.004 | -0.009 | -0.086 | 0.027 | 0.032 | 0.040 | 0.141 |
| 2 | 31.400 | 0.008 | -0.004 | -0.011 | -0.107 | 0.035 | 0.044 | 0.059 | 0.302 |
| 3 | 72.500 | 0.007 | -0.027 | -0.038 | -0.094 | 0.146 | 0.155 | 0.170 | 0.185 |
| 4 | 82.500 | 0.027 | -0.024 | -0.043 | -0.078 | 0.138 | 0.138 | 0.174 | 0.238 |
| 5 | 90.000 | 0.030 | -0.025 | -0.045 | -0.067 | 0.148 | 0.140 | 0.180 | 0.251 |
| 6 | 150.000 | -0.006 | -0.061 | -0.044 | 0.077 | 0.172 | 0.133 | 0.157 | 0.301 |

1 **Table 5**: RTTOV-gb $T_B$ uncertainty due to forward model and fast parameterization, and their total

2 squared sum for two extreme climatology conditions. Channels for the four sensors considered in the

3 current version of RTTOV-gb are given in Tables 5A (HATPRO), 5B (MP3000A), 5C (TEMPERA),

4 and 5D (LWP_K2W). Values are given for zenith observations.

| 5A - HATPRO | | | | | | |
|---|---|---|---|---|---|---|
| Chan no. (#) | Central frequency (GHz) | Fast parameterization uncertainty (K) | Absorption model uncertainty (K) | | Total uncertainty (K) | |
| | | | Tropical | Subarctic winter | Tropical | Subarctic winter |
| 1 | 22.240 | 0.037 | 0.665 | 0.290 | 0.666 | 0.292 |
| 2 | 23.040 | 0.030 | 0.621 | 0.296 | 0.621 | 0.297 |
| 3 | 23.840 | 0.026 | 0.542 | 0.303 | 0.543 | 0.304 |
| 4 | 25.440 | 0.028 | 0.480 | 0.322 | 0.481 | 0.323 |
| 5 | 26.240 | 0.027 | 0.480 | 0.332 | 0.481 | 0.333 |
| 6 | 27.840 | 0.026 | 0.506 | 0.356 | 0.506 | 0.357 |
| 7 | 31.400 | 0.030 | 0.609 | 0.420 | 0.610 | 0.421 |
| 8 | 51.260 | 0.148 | 2.623 | 3.119 | 2.628 | 3.123 |
| 9 | 52.280 | 0.167 | 2.727 | 3.301 | 2.732 | 3.305 |
| 10 | 53.860 | 0.094 | 1.003 | 1.132 | 1.007 | 1.136 |
| 11 | 54.940 | 0.024 | 0.126 | 0.089 | 0.128 | 0.093 |
| 12 | 56.660 | 0.011 | 0.023 | 0.001 | 0.026 | 0.011 |
| 13 | 57.300 | 0.009 | 0.019 | 0.003 | 0.021 | 0.009 |
| 14 | 58.000 | 0.008 | 0.018 | 0.003 | 0.020 | 0.009 |

| 5B – MP3000 | | | | | | |
|---|---|---|---|---|---|---|
| Chan no. (#) | Central frequency (GHz) | Fast parameterization uncertainty (K) | Absorption model uncertainty (K) | | Total uncertainty (K) | |
| | | | Tropical | Subarctic winter | Tropical | Subarctic winter |
| 1 | 22.234 | 0.037 | 0.665 | 0.290 | 0.666 | 0.292 |
| 2 | 22.500 | 0.036 | 0.663 | 0.292 | 0.664 | 0.294 |
| 3 | 23.034 | 0.030 | 0.621 | 0.296 | 0.622 | 0.297 |
| 4 | 23.834 | 0.026 | 0.543 | 0.303 | 0.543 | 0.304 |
| 5 | 25.000 | 0.028 | 0.487 | 0.316 | 0.487 | 0.317 |
| 6 | 26.234 | 0.027 | 0.480 | 0.332 | 0.481 | 0.333 |
| 7 | 28.000 | 0.026 | 0.509 | 0.358 | 0.510 | 0.359 |
| 8 | 30.000 | 0.028 | 0.564 | 0.393 | 0.565 | 0.394 |
| 9 | 51.248 | 0.148 | 2.619 | 3.114 | 2.624 | 3.117 |
| 10 | 51.760 | 0.157 | 2.744 | 3.299 | 2.749 | 3.302 |
| 11 | 52.280 | 0.166 | 2.727 | 3.301 | 2.732 | 3.305 |
| 12 | 52.804 | 0.165 | 2.434 | 2.943 | 2.440 | 2.948 |
| 13 | 53.336 | 0.141 | 1.793 | 2.129 | 1.798 | 2.134 |
| 14 | 53.848 | 0.094 | 1.020 | 1.153 | 1.024 | 1.156 |
| 15 | 54.400 | 0.046 | 0.390 | 0.388 | 0.393 | 0.391 |
| 16 | 54.940 | 0.024 | 0.126 | 0.089 | 0.128 | 0.093 |
| 17 | 55.500 | 0.016 | 0.052 | 0.018 | 0.054 | 0.024 |
| 18 | 56.020 | 0.013 | 0.033 | 0.004 | 0.035 | 0.014 |
| 19 | 56.660 | 0.011 | 0.023 | 0.001 | 0.026 | 0.011 |

| | | | | | | |
|---|---|---|---|---|---|---|
| 20 | 57.288 | 0.009 | 0.019 | 0.003 | 0.021 | 0.009 |
| 21 | 57.964 | 0.008 | 0.018 | 0.003 | 0.020 | 0.009 |
| 22 | 58.800 | 0.007 | 0.018 | 0.004 | 0.019 | 0.008 |

| 5C - TEMPERA | | | | | | |
|---|---|---|---|---|---|---|
| Chan no. (#) | Central frequency (GHz) | Fast parameterization uncertainty (K) | Absorption model uncertainty (K) | | Total uncertainty (K) | |
| | | | Tropical | Subarctic winter | Tropical | Subarctic winter |
| 1 | 51.250 | 0.148 | 2.620 | 3.115 | 2.624 | 3.118 |
| 2 | 51.750 | 0.157 | 2.743 | 3.296 | 2.747 | 3.300 |
| 3 | 52.250 | 0.166 | 2.733 | 3.307 | 2.738 | 3.311 |
| 4 | 52.850 | 0.164 | 2.393 | 2.892 | 2.398 | 2.896 |
| 5 | 53.350 | 0.141 | 1.773 | 2.104 | 1.778 | 2.109 |
| 6 | 53.850 | 0.094 | 1.017 | 1.149 | 1.021 | 1.153 |
| 7 | 54.400 | 0.046 | 0.390 | 0.388 | 0.393 | 0.391 |
| 8 | 54.900 | 0.024 | 0.136 | 0.100 | 0.138 | 0.103 |
| 9 | 55.400 | 0.017 | 0.059 | 0.023 | 0.061 | 0.029 |
| 10 | 56.000 | 0.013 | 0.033 | 0.004 | 0.036 | 0.014 |
| 11 | 56.500 | 0.011 | 0.025 | 0.000 | 0.027 | 0.011 |
| 12 | 57.000 | 0.010 | 0.021 | 0.002 | 0.023 | 0.010 |

| 5D – LWP_K2W | | | | | | |
|---|---|---|---|---|---|---|
| Chan no. (#) | Central frequency (GHz) | Fast parameterization uncertainty (K) | Absorption model uncertainty (K) | | Total uncertainty (K) | |
| | | | Tropical | Subarctic winter | Tropical | Subarctic winter |
| 1 | 23.840 | 0.026 | 0.542 | 0.303 | 0.543 | 0.304 |
| 2 | 31.400 | 0.030 | 0.609 | 0.420 | 0.610 | 0.421 |
| 3 | 72.500 | 0.139 | 2.775 | 3.690 | 2.778 | 3.692 |
| 4 | 82.500 | 0.119 | 2.706 | 2.042 | 2.708 | 2.045 |
| 5 | 90.000 | 0.126 | 2.963 | 1.665 | 2.966 | 1.669 |
| 6 | 150.000 | 0.161 | 3.547 | 2.118 | 3.550 | 2.124 |

4

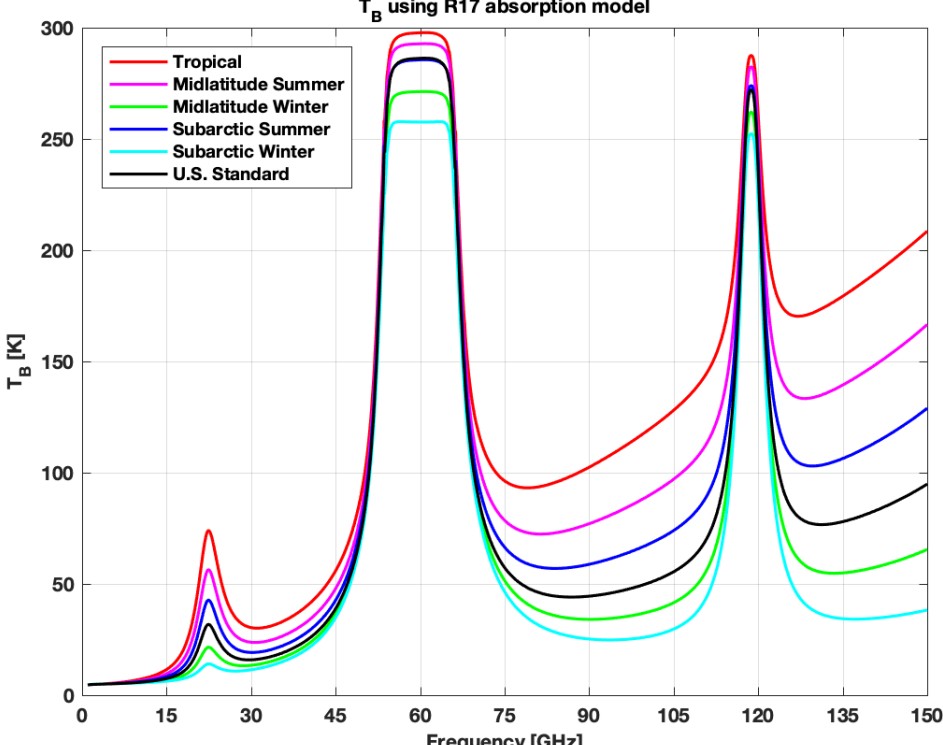

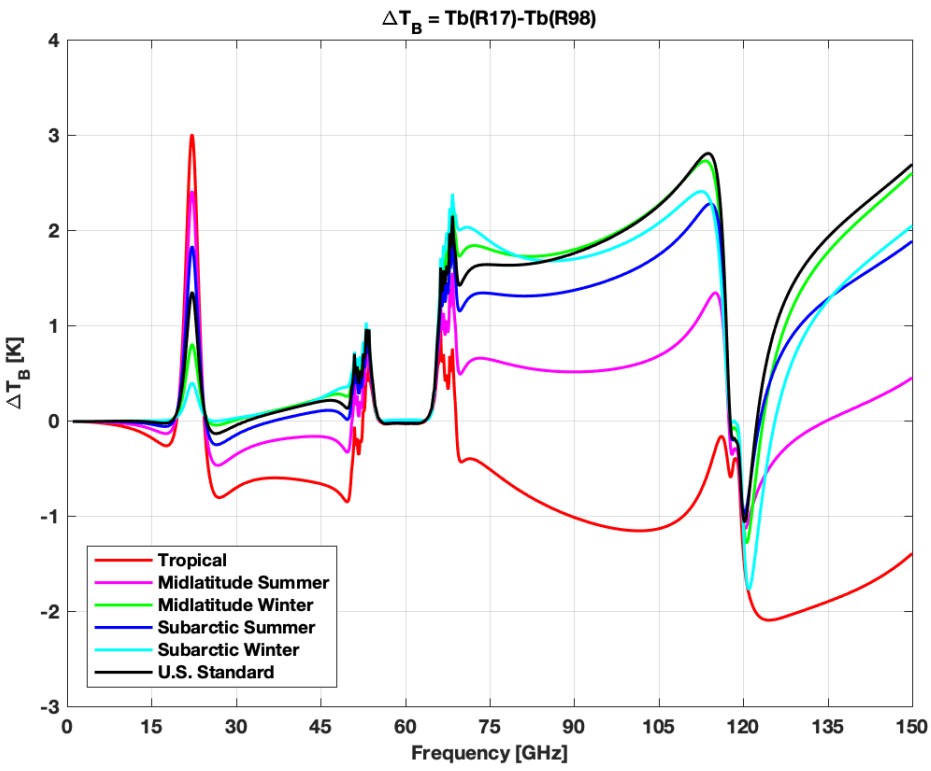

3    **Figure 1: (Top) Zenith downwelling $T_B$ computed using six reference atmosphere climatology conditions with the R17**
4    **model. (Bottom) Difference between $T_B$ computed with the current and reference versions (R17 minus R98) for the six**
5    **atmosphere climatology conditions. This figure is similar to Figure 1 in Cimini et al. (2018), although $T_B$ were**
6    **recomputed to cover a wider frequency range.**

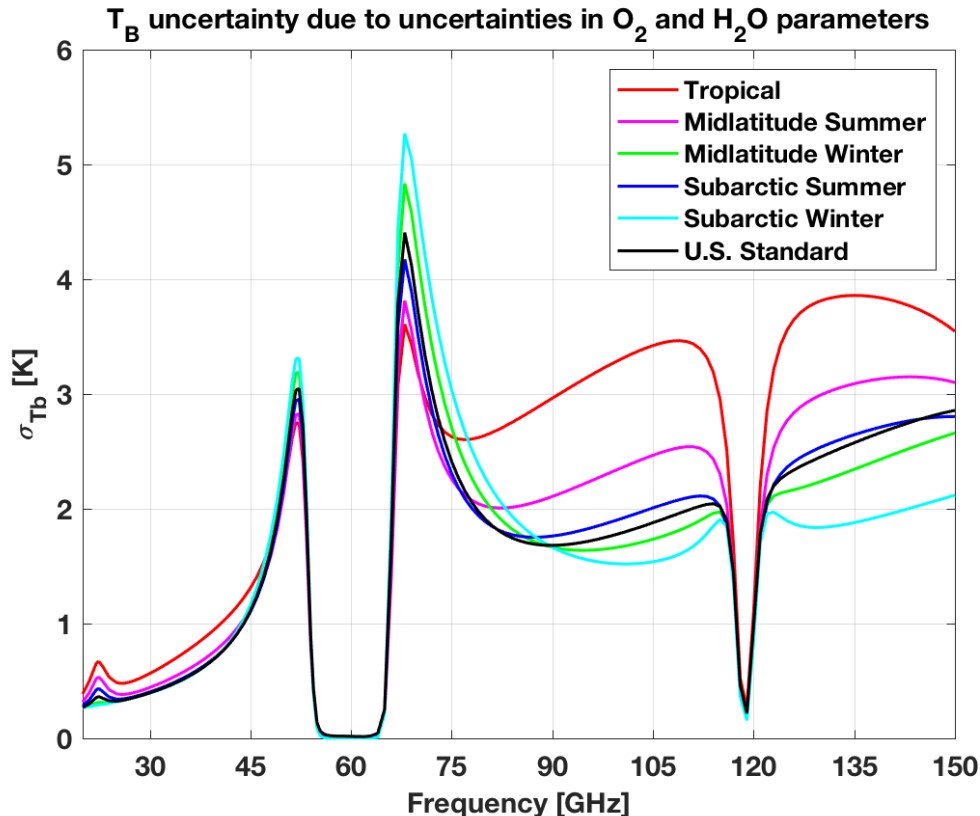

Figure 2: Zenith downwelling $T_B$ uncertainty ($\sigma(T_B)$) due to the uncertainty in $O_2$ and $H_2O$ absorption model parameters. Six climatological atmospheric conditions (color-coded) have been used to compute $Cov(p)$ (see Section 2.2). $\sigma(T_B)$ is computed as the square root of the diagonal terms of $Cov(T_B)$. This figure is similar to Figure 6 in Cimini et al. (2018), although $\sigma(T_B)$ was recomputed to cover a wider frequency range.

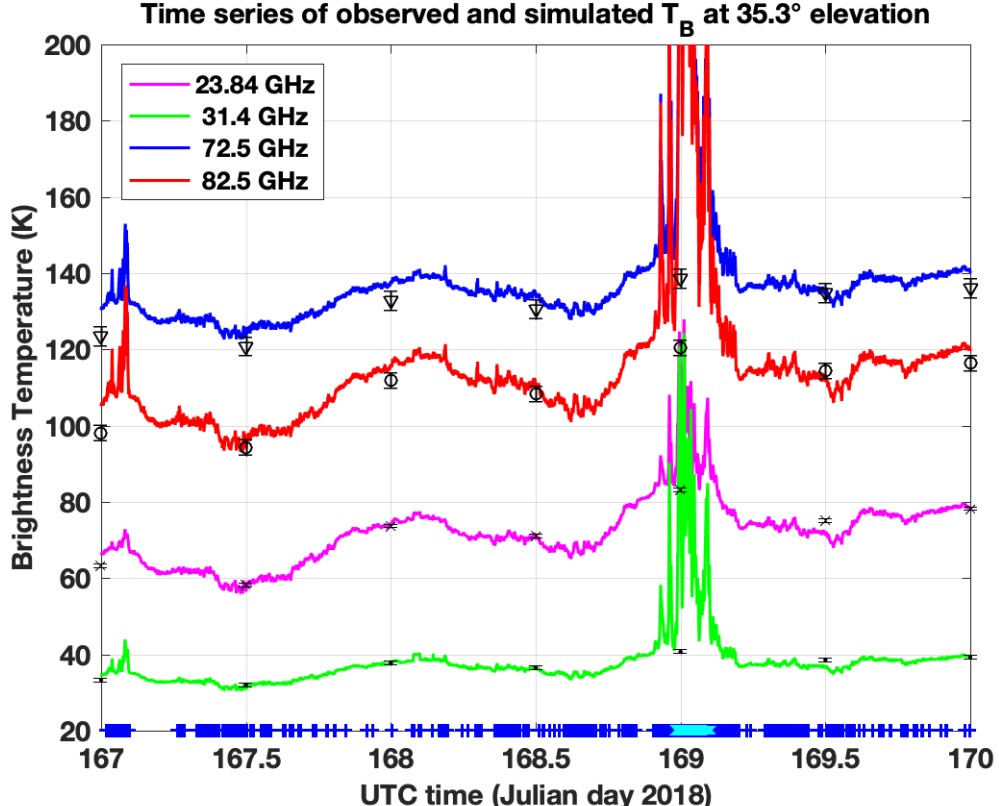

Figure 3: Time series of observed (lines) and simulated (markers) $T_B$ at 35.3° elevation for four channels of LWP-U72-82. The radiometer is located at the Polytechnic University campus in Milan (Italy), while radiosondes used for the simulations are launched from the Milan Linate airport (~20 km from the Politechnic University campus). Channel frequencies are color-coded as reported in the legend. Simulations are reported with dots (23.84 GHz), crosses (31.4 GHz), triangles (72.5 GHz), and circles (82.5 GHz), including an indicative estimate of the total uncertainty. The cloud and rain flags are indicated at the bottom by blue and cyan crosses, respectively. The time series spans from 00:00 of 16 June (Julian day 167) to 00:00 of 19 June (Julian day 170) 2018.

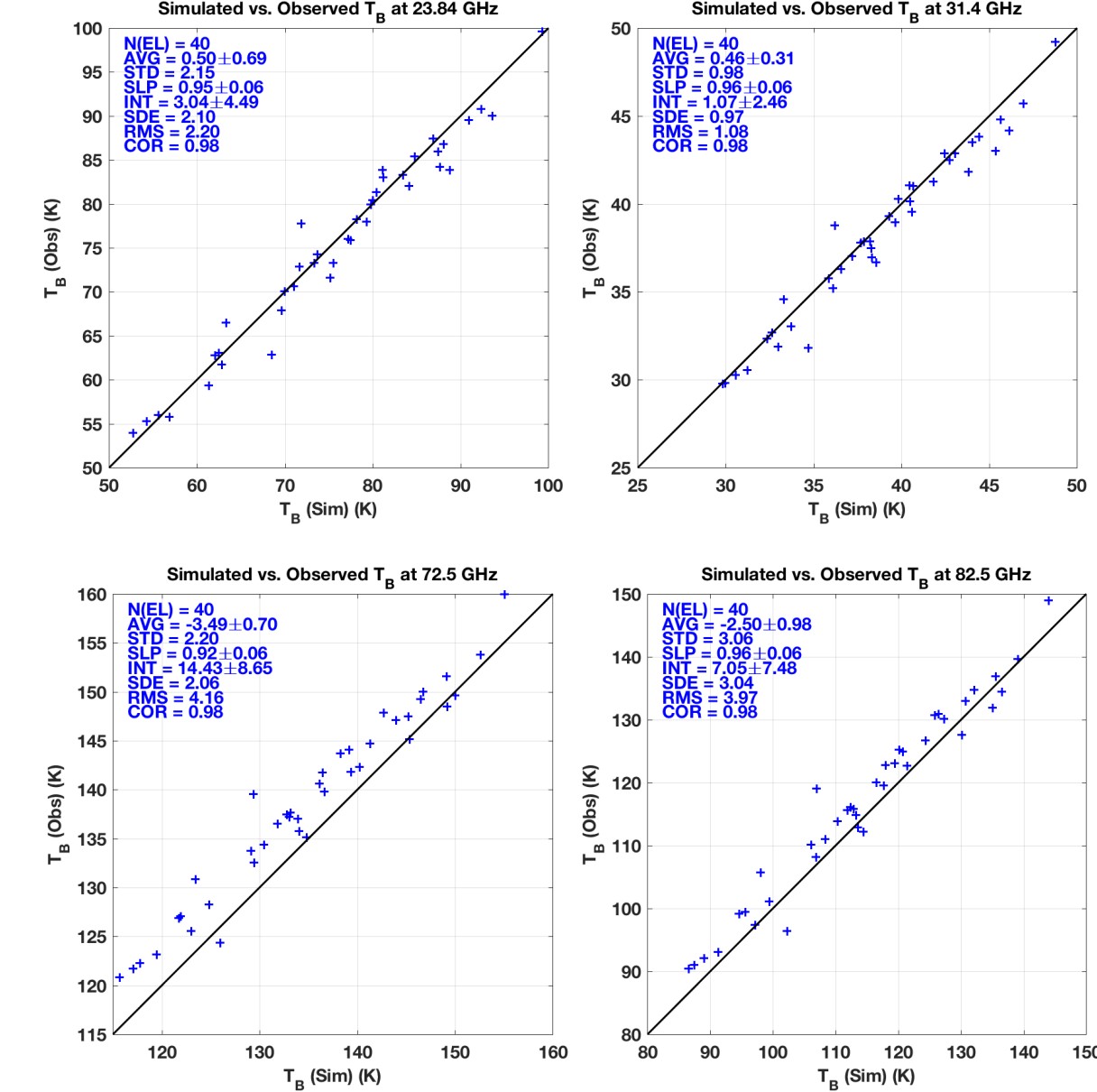

Figure 4: Scatter of simulated vs. observed T_B at 35.3° elevation for four channels of LWP-U72-82. Location of radiometer and radiosondes are as in Figure 3. The absorption model of Rosenkranz 2017 has been used. Each panel reports the number of elements (N(EL)), the average difference (AVG), the standard deviation (STD), the slope (SLP) and intercept (INT) of a linear fit, the standard error (SDE), the root-mean-square (RMS), and correlation coefficient (COR). 95% confidence intervals are given for AVG, SLP, and INT. Units for AVG, STD, SDE, and RMS are Kelvin.

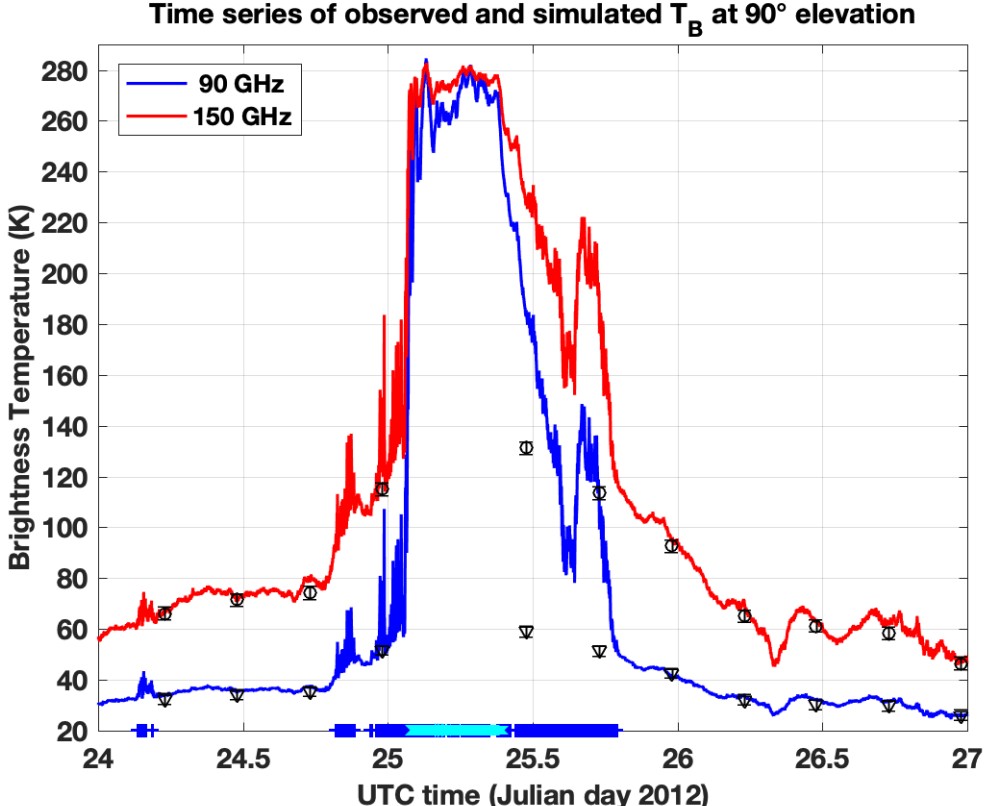

Figure 5: Time series of observed (lines) and simulated (markers) $T_B$ at 90° elevation for two channels of LWP-90-150. The radiometer and radiosondes are operated from the Atmospheric Radiation Measurement (ARM) program Southern Great Plains (SGP) central facility (Lamont, OK, USA). Channel frequencies are color-coded as reported in the legend. Simulations are reported with triangles (90 GHz) and circles (150 GHz), including an indicative estimate of the total uncertainty. The cloud and rain flags are indicated at the bottom by blue and cyan crosses, respectively. The time series spans from Jan 24 00:00 to Jan 27 00:00 UTC 2012 (Julian day 24-27).

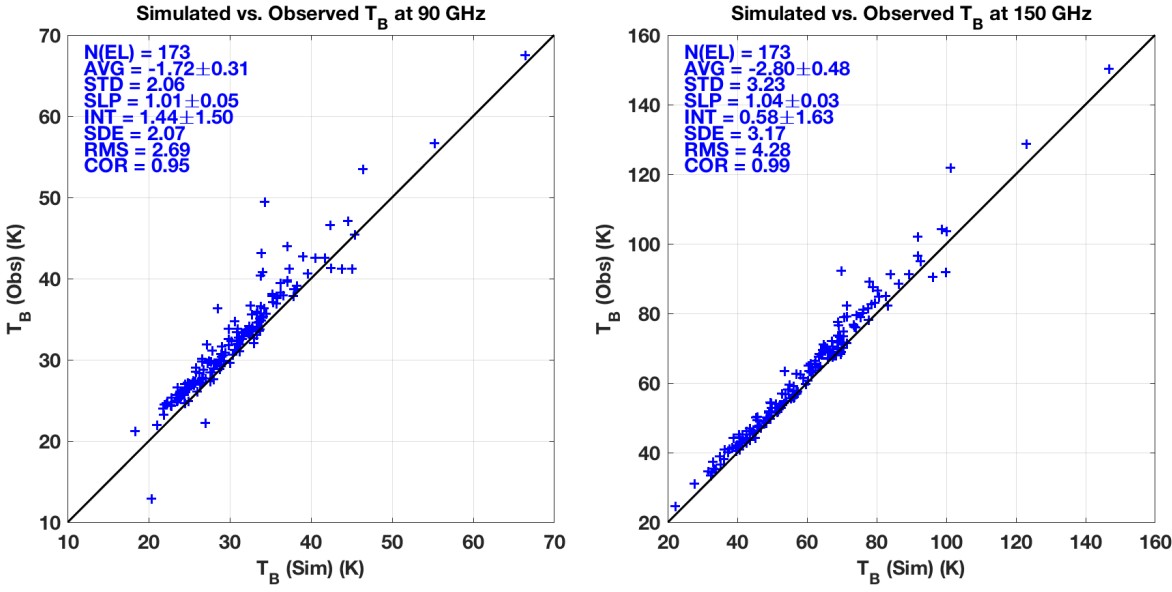

Figure 6: Same as Figure 4 but showing simulated vs. observed $T_B$ at 90° elevation for two channels of LWP-90-150. Location of radiometer and radiosondes are as in Figure 5. The absorption model of Rosenkranz 2017 has been used.

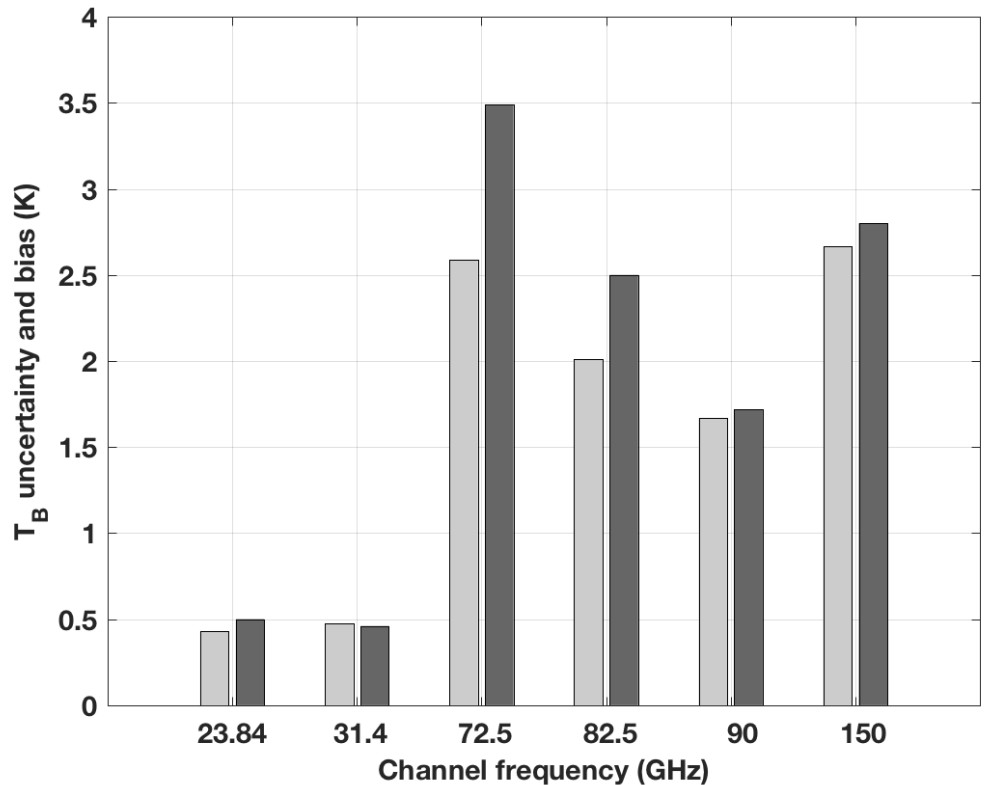

Figure 7: Estimated uncertainty (light grey) and experimental mean difference (dark grey) for six LWP_K2W channels. Radiometric observations were collected in June-July in Milan (45°N) with the four lower frequency channels, while in January-February in Lamont (36°N) with the two higher frequency channels. Thus, uncertainty is estimated using midlatitude summer conditions for the four lower frequency channels, while midlatitude winter conditions for the two higher frequency channels.