# Peer review of "RTTOV-gb v1.0 - Updates on sensors, absorption models, 1"

_Geoscientific Model Development, 2018_

## Referee Comment (RC1) · Anonymous Referee #1 · 13 Feb 2019

General comments:

The manuscript presents an updated version of a fast radiative transfer model (RTTOV-gb) for ground-based microwave radiometers. It discusses model uncertainties caused by absorption properties in the microwave range and shows validations of the model with observations.

The paper wants to contribute to a wider use of ground-based microwave brightness temperatures to be assimilated in numerical weather prediction models which has been shown to have a promising impact to the forecast skill.

The paper is written in a clear and concise way, and I suggest publishing with minor revisions.

[Figure]

Specific comments:

Why did you restrict yourself to the frequency range 22-150 GHz? Did you also check the 183 GHz water vapor line which would be of great use in arctic or high altitude regions with low water vapor contents?

Is there any attempt to take cloud liquid water into account for RTTOV-gb? Please comment on that in the paper since there are many regions on Earth with frequent cloud cover.

p.3, l.6-8: I assume the coefficients are instrument type specific. This would mean that you can use the same coefficients for all stations with the same instrument type in different climate zones. Is that correct? That was not entirely clear to me.

p.5 and 6: Concerning the comparison of the model with observations: Did you check the calibrations of the instruments? Were there any absolute calibrations performed during the periods of study?

Technical corrections:

p. 1, l.26: Therefore (typo)

p.1, l.36: To my mind "model flavor" sounds a bit colloquial. Could you find a better expression?

p.2, l.11: not only "national" meteorological services (e.g. ECMWF!)

p.3, l.36: Turner et al. 2018 has not been submitted yet

p.4, l.1: Tretyakov is misspelled

---

## Author Comment (AC1) · 22 Mar 2019

**General comments:**

The manuscript presents an updated version of a fast radiative transfer model (RTTOV-gb) for ground-based microwave radiometers. It discusses model uncertainties caused by absorption properties in the microwave range and shows validations of the model with observations. The paper wants to contribute to a wider use of ground-based microwave brightness temperatures to be assimilated in numerical weather prediction models which has been shown to have a promising impact to the forecast skill. The paper is written in a clear and concise way, and I suggest publishing with minor revisions.

We thank the reviewer for her/his careful reading and positive comments.

[Figure]

**Specific comments:**

Why did you restrict yourself to the frequency range 22-150 GHz? Did you also check the 183 GHz water vapor line which would be of great use in arctic or high altitude regions with low water vapor contents?

The frequency range depends on the popularity of ground-based microwave radiometer types. In the first paper introducing RTTOV-gb (De Angelis et al., 2016 – references are reported at the end of this document), we only considered 22-60 GHz channels, addressing the majority of operational ground-based microwave radiometers (i.e. RPG HATPRO and Radiometrics MP3000). Here, we add new instruments, extending the frequency range up to 150 GHz, since such types are becoming popular in support to satellite telecommunications. We are aware of ground-based microwave radiometers operating at 183 GHz and their benefit for low water vapor content (e.g. Cimini et al., 2007; Cadeddu et al., 2007; Ricaud et al., 2010). However, only few operational units are available in the world. The extension to 183 GHz will be considered among future developments. We have added this comment to Sections 2.1 and 4. Accordingly, the latter has been renamed "Summary and future developments".

Is there any attempt to take cloud liquid water into account for RTTOV-gb? Please comment on that in the paper since there are many regions on Earth with frequent cloud cover.

Thanks for pointing this out. As explained in the first paper (De Angelis et al., 2016), RTTOV-gb does already include cloud liquid water in the radiative transfer calculations (direct and Jacobian modules). Here, we showed clear-sky calculations only for the purpose of comparing with operational radiosonde observations, which do not provide measurements of cloud liquid water content. We have added text to clarify this point to Sections 1 and 3.

p.3, l.6-8: I assume the coefficients are instrument type specific. This would mean that you can use the same coefficients for all stations with the same instrument type in

different climate zones. Is that correct? That was not entirely clear to me.

Yes, that's correct. The coefficient training is based on a set of diverse profiles which covers the atmospheric conditions of different climate zones. We have added text to clarify this point in Section 2.1.

p.5 and 6: Concerning the comparison of the model with observations: Did you check the calibrations of the instruments? Were there any absolute calibrations performed during the periods of study?

For the radiometer in Milan (LWP-U72-82), the most recent absolute calibration was performed 9 months earlier than the period of study. A new calibration was performed 4 months later, and did not show substantial changes in the calibration coefficients. Thus, we assume the calibration was stable during the period of study. The radiometer in Lamont (LWP-90-150) is continuously calibrated using the tip curve method, as regularly performed by ARM (Cadeddu et al., 2013). We have added text to clarify this point to Section 3.

**Technical corrections:**

p. 1, l.26: Therefore (typo)

Addressed. Thanks!

p.1, l.36: To my mind "model flavor" sounds a bit colloquial. Could you find a better expression?

Agreed. It has been changed with "model parameterization".

p.2, l.11: not only "national" meteorological services (e.g. ECMWF!)

Correct. It has been changed with "national and international". Thanks!

p.3, l.36: Turner et al. 2018 has not been submitted yet

It has been now replaced with the proper reference to the published paper.

p.4, l.1: Tretyakov is misspelled

Addressed. Thanks!

**References**

Cadeddu et al., 2007, doi: 10.1109/MICRAD.2006.1677098
Cadeddu et al., 2013, doi: 10.5194/amt-6-2359-2013
Cimini et al., 2007, doi: 10.1109/TGRS.2007.897450
De Angelis et al., 2016, doi:10.5194/amt-10-3947-2017
Ricaud et al., 2010, doi: 10.1109/TGRS.2009.2029345
Turner et al., 2019, doi: 10.1016/j.jqsrt.2019.02.013

––––––––––––––––––––––––––––––

---

## Referee Comment (RC2) · Anonymous Referee #2 · 25 Mar 2019

**1  General comments**

The authors describe the atmospheric radiative transfer code RTTOV adapted for ground-based sensors (RTTOV-gb) and present some updates compared to earlier studies (e.g. De Angelis et al., 2016), by adding two more sensors and presenting model uncertainties. They test the RTTOV-gb model by comparing simulated brightness temperatures ($T_B$) with a full line-by-line (LBL) model. Further, they compare $T_B$ simulated from radiosonde data with measured $T_B$ from co-located microwave radiometers. This approach provides a complete vaildation of the model, using both, the reference model and measured data as comparison. They justify the importance of the RTTOV-gb by its use to assimilate groundbased microwave radiometer data in NWP

models.

The paper is well written and methods and results are presented in a concise way. It presents model advances with the aim to improve NWP and fits well in the scope of GMD. I recommend to publish the manuscript in GMD after consideration of my minor comments given below.

**2 Specific comments**

p. 1, l. 23: "... computes the bottom of atmosphere radiances". What do you mean with "bottom"? Do you mean that RTTOV-gb computes radiances and brightness temperatures of the lower atmosphere/troposphere?

p. 2 , l. 4-6: Please put the first sentence of the introduction ("RTTOV-gb is the FOR-TRAN....") after the second sentence ("RTTOV-gb is a fast .... (i.e. radiances)"). The latter is a very nice introductory sentence, whereas the first gives additional information and should therefore not be the first sentence of the manuscript.

p.2, l.28: you mention that RTTOV-gb is one-dimensional: clarify which dimension is meant, e.g. add " profiles .... at a specific location".

p. 2, l. 36-37: "added as options" is not clear.

p. 2, l. 37: evaluation of what? Add e.g. "evaluation of RTTOV-gb against ....".

p. 3, l. 5: I suggest to place the second paragraph (p. 3, l. 15) after the first sentence in line 5. This would fist generally describe the new sensors and thus provide a direct link to the title of the section (New sensors). Afterwards, the it would describe the more specific information concerning the parametrization and the coefficients.

p. 3, l. 10: Reference to Sect. 2.2, where the regression coefficients are explained in more detail.

p. 3, l. 18: Mention also the study of Stähli et al. 2013 (doi:10.5194/amt-6-2477-2013), which first described the TEMPERA instrument.

p. 3, l. 20: "LWP family": mention the sensors from p. 5, l. 23-24 already here, because this is the section where you describe the new sensors.

p. 3, l. 22: Please add some more information about the channels of the sensors used, or reference to Table 2 not only for the supported sensors, but also for their channels.

p. 4, l. 8: Clarify that it is not the first study that uses both models. E.g. "both ... models are now available and were first presented in Cimini et al. (2018). Extending the results ...".

p.4, l. 21-22: I suggest to move these 2 sentences to p.4, l. 8, where Cimini et al. and the broader frequency range of the present study has been mentioned first.

p. 4, l. 23-24: "water vapor self-boradened continuum temperature dependence exponent $n_{cs}$": The long noun cluster is confusing. I prefer a description similar to Cimini et al. (2018), e.g. "... the temperature-dependence exponent $n_{cs}$ of the water vapor self-broadened continuum ..."

p. 4, l. 32: Section 3 is rather long compared to the other sections. I suggest to divide the section into two subsections, one about the comparison with the reference LBL model (starting in p.4, l. 34) and the second one about the comparison with the real observations (starting in p. 5, l. 19).

p. 5, l. 4: What does "main differences" mean? Please remove "main".

p. 5, l. 5: Add "generally" before "decrease" and "increase", because this behaviour is not valid for all of the channels (e.g. Table 3, ch. 1 first increases, and Table 5, ch. 5 first decreases).

p. 5, l. 5: Add Table reference or sensor type for clarity, e.g. "..., the rms differences GENERALLY decrease for 50-57 GHz channels (TEMPERA, Table 3), while they increase for 23/31 and 70-150 GHz channels (LWP, Table 4)".

p. 5, l. 8: add "(RTTOV-gb)" to "fast model approximation" to clarify which model is meant.

p. 5, l. 5: Which angle is used for this "fast parametrization uncertainty" in Table 5?

p. 5, l. 19: Begin a new section (see comment above, p. 4, l. 32).

p. 5, l. 27: Please mention already here that these datasets cover two different meteorological conditions, namely midlatitude summer and midlatitude winter conditions.

p. 5, l. 34: Which altitude limits for the calculations do you generally use? Do the radiosondes cover this whole altitude range?

p. 6, l. 4: Why did you choose a threshold of 0.2K when reducing the 0.5K threshold from 1h to 10 minutes?

p. 6, l. 5: It would be nice to mark these identified cloudy periods in Figure 3 and Figure 5.

p. 6, l. 11: Could you provide some more details? What do you mean with conditions-dependent uncertainties?

p. 6, l. 18: It is interesting to mention here the meteorological conditions for the dataset. Please mention earlier also the conditions for the Milan dataset (see comment above, p. 5, l. 27).

p. 7, l. 5: Besides in the manuscript title and the Abstract, it is the first time here that you mention the version (v1.0) of RTTOV-gb, and you continue to do so during the whole summary section. Please mention the version earlier in the text and be consistent in the usage.

p. 7, l. 16: I think you can be more confident here. I suggest to change it to "This paper can provide ...".

[Figure]

p. 8, l. 20: For completeness, also state where RS98 is available.

p. 17, Fig. 1: Adapt figure titles ("ros17" and "rosen") to R17 and R98 to be consistent with the text.

p. 18, l. 3: $K_p$ is not defined, do you mean **Cov(p)**?

p. 19, Fig. 3: Add unit (GHz) to the legend. Also, it would be nice to mark cloudy periods (see comment p. 6, l. 5).

p. 19, l. 3: add "radiosondes used for the simulations ...". Otherwise it is not clear why you mention radiosondes here in the caption.

p. 20, Fig. 4: the x-labels of the first two panels are cut.

p. 21, Fig. 5: Same comment as for Fig. 3, add unit to legend and mark cloudy periods.

**3   Technical corrections**

p. 1, l. 33: Typo, according to the tables it should be 82.5.

p. 2, l. 22: Data assimilation (DA): abbreviation not needed, because it is not used later on.

p. 2, l. 31: "As hoped" is not appropriate here, state it as a fact.

p. 2, l. 38: "Section 5 provideS"

p. 3, l. 19: "Liquid Water Path (LWP)". Introduce the abbreviation LWP.

p. 3, l. 35: Provide definition of AMSUTRAN.

p.4, l. 7: remove blank space before ")".

p. 4, l. 25: "the same approach AS described in ..."

p. 4, l. 27: bold typesetting of **Cov(...)** and **Cor(...)** for consistency.

p. 6, l. 2: Missing space after T$_B$

p. 6, l. 5: Missing space after T$_B$

p. 6, l. 22: Missing space after T$_B$

p. 6, l. 24: Missing space after T$_B$

p. 12, l. 1: Add "with the corresponding sensor channels (sensor chans)."

p. 13, Table 3: Add a digit to the Central frequencies to be consistent with the number of digits in the other columns and in Table 5.

p. 14, Table 4: Same as for Table 3, add a digit to the channel frequencies.

p. 15, Table 5: Add units for the uncertainties.

p. 20, l. 7: add "and": AVG, STD, SDE, and RMS

---

## Author Comment (AC2) · 30 Mar 2019

**General comments:**

The authors describe the atmospheric radiative transfer code RTTOV adapted for ground-based sensors (RTTOV-gb) and present some updates compared to earlier studies (e.g. De Angelis et al., 2016), by adding two more sensors and presenting model uncertainties. They test the RTTOV-gb model by comparing simulated brightness temperatures (TB) with a full line-by-line (LBL) model. Further, they compare TB simulated from radiosonde data with measured TB from co-located microwave radiometers. This approach provides a complete vaildation of the model, using both, the reference model and measured data as comparison. They justify the importance

of the RTTOV-gb by its use to assimilate groundbased microwave radiometer data in NWP models. The paper is well written and methods and results are presented in a concise way. It presents model advances with the aim to improve NWP and fits well in the scope of GMD. I recommend to publish the manuscript in GMD after consideration of my minor comments given below.

We thank the reviewer for her/his careful reading and positive comments.

**Specific comments:**

p. 1, l. 23: "... computes the bottom of atmosphere radiances". What do you mean with "bottom"? Do you mean that RTTOV-gb computes radiances and brightness temperatures of the lower atmosphere/troposphere?

No. We mean that RTTOV-gb computes the downwelling radiances (and brightness temperatures) leaving the bottom of the atmosphere. The radiative transfer extends to the top of the atmosphere and it does include extra-terrestrial contribution, i.e. the cosmic background radiation. We rephrased the abstract and have added text to Section 1 to make it more clear.

p. 2 , l. 4-6: Please put the first sentence of the introduction ("RTTOV-gb is the FOR-TRAN....") after the second sentence ("RTTOV-gb is a fast .... (i.e. radiances)"). The latter is a very nice introductory sentence, whereas the first gives additional information and should therefore not be the first sentence of the manuscript.

Agreed.

p.2, l.28: you mention that RTTOV-gb is one-dimensional: clarify which dimension is meant, e.g. add " profiles .... at a specific location".

Agreed.

p. 2, l. 36-37: "added as options" is not clear.

Agreed. We replaced with "added among the setting options".

p. 2, l. 37: evaluation of what? Add e.g. "evaluation of RTTOV-gb against ....".

Agreed.

p. 3, l. 5: I suggest to place the second paragraph (p. 3, l. 15) after the first sentence in line 5. This would fist generally describe the new sensors and thus provide a direct link to the title of the section (New sensors). Afterwards, the it would describe the more specific information concerning the parametrization and the coefficients.

Agreed.

p. 3, l. 10: Reference to Sect. 2.2, where the regression coefficients are explained in more detail.

Agreed.

p. 3, l. 18: Mention also the study of Stahli et al. 2013 (doi:10.5194/amt-6-2477-2013), which first described the TEMPERA instrument.

Agreed.

p. 3, l. 20: "LWP family": mention the sensors from p. 5, l. 23-24 already here, because this is the section where you describe the new sensors.

Agreed.

p. 3, l. 22: Please add some more information about the channels of the sensors used, or reference to Table 2 not only for the supported sensors, but also for their channels.

Agreed.

p. 4, l. 8: Clarify that it is not the first study that uses both models. E.g. "both ... models are now available and were first presented in Cimini et al. (2018). Extending the results ...".

We mean that both models are now available within RTTOV-gb. This was not true at the time of Cimini et al. (2018). We added text to make it clear.

p.4, l. 21-22: I suggest to move these 2 sentences to p.4, l. 8, where Cimini et al. and the broader frequency range of the present study has been mentioned first.

We see the reviewer's point. But the 2 sentences at lines 21-22 assume the reader is aware of the sensitivity study approach (summarized in lines 12-20) and thus should follow the summary. Therefore, we prefer to keep the current order.

p. 4, l. 23-24: "water vapor self-boradened continuum temperature dependence exponent ncs": The long noun cluster is confusing. I prefer a description similar to Cimini et al. (2018), e.g. "... the temperature-dependence exponent ncs of the water vapor self-broadened continuum ..."

Agreed.

p. 4, l. 32: Section 3 is rather long compared to the other sections. I suggest to divide the section into two subsections, one about the comparison with the reference LBL model (starting in p.4, l. 34) and the second one about the comparison with the real observations (starting in p. 5, l. 19).

Agreed.

p. 5, l. 4: What does "main differences" mean? Please remove "main".

Agreed.

p. 5, l. 5: Add "generally" before "decrease" and "increase", because this behaviour is not valid for all of the channels (e.g. Table 3, ch. 1 first increases, and Table 5, ch. 5 first decreases).
p. 5, l. 5: Add Table reference or sensor type for clarity, e.g. "..., the rms differences GENERALLY decrease for 50-57 GHz channels (TEMPERA, Table 3), while they increase for 23/31 and 70-150 GHz channels (LWP, Table 4)".

Both agreed.

p. 5, l. 8: add "(RTTOV-gb)" to "fast model approximation" to clarify which model is meant.

Agreed.

p. 5, l. 5: Which angle is used for this "fast parametrization uncertainty" in Table 5?

As specified in the caption of Table 5, the considered angle is Zenith (i.e. 90° elevation). We have now added this information within the text as well.

p. 5, l. 19: Begin a new section (see comment above, p. 4, l. 32).

Agreed.

p. 5, l. 27: Please mention already here that these datasets cover two different meteorological conditions, namely midlatitude summer and midlatitude winter conditions.

Agreed.

p. 5, l. 34: Which altitude limits for the calculations do you generally use? Do the radiosondes cover this whole altitude range?

The pressure limits are given in Table 1, which cover from surface to the top of the atmosphere. Radiosondes usually reach up to 10 hPa ( 30 km altitude), leaving the uppermost 5 levels to be covered with climatological profiles. This has negligible impact on ground-based radiance calculations. We added text in Section 3.2 to clarify.

p. 6, l. 4: Why did you choose a threshold of 0.2K when reducing the 0.5K threshold from 1h to 10 minutes?

We assume that the clear-sky atmospheric variability decreases with decreasing time interval. By plotting sigma(Tb(30GHz))@10m in increasing order, we notice a steady slope up to 0.2K (which we assume is due to the clear-sky variability) followed by a rapid increase (associated to cloud contamination). Thus, we set 0.2K as the threshold

for sigma(Tb(30GHz))@10m to indicate clear-sky conditions. We rephrased the text to clarify.

p. 6, l. 5: It would be nice to mark these identified cloudy periods in Figure 3 and Figure 5.

Agreed.

p. 6, l. 11: Could you provide some more details? What do you mean with conditions-dependent uncertainties?

The 72.5 GHz channel is influenced by both temperature and water vapor, and thus its uncertainty. Figure 2 suggests that the uncertainty at 72.5 GHz increases as temperature decreases, while it decreases as moisture increases. Thus, the different slope in Figure 4 may be partially due to increasing uncertainty at lower temperature and moisture conditions. We added text in Section 3.2 to clarify.

p. 6, l. 18: It is interesting to mention here the meteorological conditions for the dataset. Please mention earlier also the conditions for the Milan dataset (see comment above, p. 5, l. 27).

Agreed.

p. 7, l. 5: Besides in the manuscript title and the Abstract, it is the first time here that you mention the version (v1.0) of RTTOV-gb, and you continue to do so during the whole summary section. Please mention the version earlier in the text and be consistent in the usage.

Agreed. We have added the version number where deemed appropriate.

p. 7, l. 16: I think you can be more confident here. I suggest to change it to "This paper can provide ...".

Agreed. We changed into "We expect this paper will provide. . ."

**GMDD**

p. 8, l. 20: For completeness, also state where RS98 is available.

The R98 code is no longer supported and thus it is not available through a repository.

p. 17, Fig. 1: Adapt figure titles ("ros17" and "rosen") to R17 and R98 to be consistent with the text.

Agreed.

p. 18, l. 3: Kp is not defined, do you mean Cov(p)?

Correct. We modified the caption accordingly. Thanks much for spotting this typo..

p. 19, Fig. 3: Add unit (GHz) to the legend. Also, it would be nice to mark cloudy periods (see comment p. 6, l. 5).

Agreed.

p. 19, l. 3: add "radiosondes used for the simulations ...". Otherwise it is not clear why you mention radiosondes here in the caption.

Agreed.

p. 20, Fig. 4: the x-labels of the first two panels are cut.

Addressed. Thanks.

p. 21, Fig. 5: Same comment as for Fig. 3, add unit to legend and mark cloudy periods.

Agreed.

**Technical corrections:**

p. 1, l. 33: Typo, according to the tables it should be 82.5.
p. 2, l. 22: Data assimilation (DA): abbreviation not needed, because it is not used later on.
p. 2, l. 31: "As hoped" is not appropriate here, state it as a fact.

p. 2, l. 38: "Section 5 provideS"

p. 3, l. 19: "Liquid Water Path (LWP)". Introduce the abbreviation LWP.

All the above are agreed. Thanks!

p. 3, l. 35: Provide definition of AMSUTRAN.

Since AMSUTRAN is not used here, we prefer to refer to the original publication (Turner et al., 2019) for definition and further details.

p.4, l. 7: remove blank space before ")".

p. 4, l. 25: "the same approach AS described in ..."

p. 4, l. 27: bold typesetting of Cov(...) and Cor(...) for consistency.

All the above are agreed.

p. 6, l. 2: Missing space after TB

p. 6, l. 5: Missing space after TB

p. 6, l. 22: Missing space after TB

p. 6, l. 24: Missing space after TB

With TB(90GHz) we intend TB at 90 GHz, so we prefer to keep this compact notation.

p. 12, l. 1: Add "with the corresponding sensor channels (sensor chans)."

p. 13, Table 3: Add a digit to the Central frequencies to be consistent with the number of digits in the other columns and in Table 5.

p. 14, Table 4: Same as for Table 3, add a digit to the channel frequencies.

p. 15, Table 5: Add units for the uncertainties.

p. 20, l. 7: add "and": AVG, STD, SDE, and RMS

All the above are agreed. Thanks!